# Nucleotide Imbalance, Provoked by Downregulation of Aspartate Transcarbamoylase Impairs Cold Acclimation in Arabidopsis

**DOI:** 10.3390/molecules28041585

**Published:** 2023-02-07

**Authors:** Leo Bellin, Diana Laura Garza Amaya, Vanessa Scherer, Tobias Pruß, Annalisa John, Andreas Richter, Torsten Möhlmann

**Affiliations:** 1Faculty of Biology, Plant Physiology, University of Kaiserslautern, Erwin-Schrödinger-Straße, D-67663 Kaiserslautern, Germany; 2Institute for Biosciences, Physiology of Plant Metabolism, University of Rostock, Albert-Einstein-Strasse 3, D-18059 Rostock, Germany

**Keywords:** pyrimidine de novo synthesis, cold acclimation, transcriptomics

## Abstract

Aspartate transcarbamoylase (*ATC*) catalyzes the first committed step in pyrimidine de novo synthesis. As shown before, mutants with 80% reduced transcript and protein levels exhibit reduced levels of pyrimidine metabolites and thus nucleotide limitation and imbalance. Consequently, reduced photosynthetic capacity and growth, accompanied by massive transcriptional changes, were observed. Here, we show that nucleotide de novo synthesis was upregulated during cold acclimation of *Arabidopsis thaliana* (ecotype Columbia, Col-0) plants, but *ATC* knockdown mutants failed to acclimate to this condition as they did not accumulate neutral sugars and anthocyanins. A global transcriptome analysis revealed that most of the transcriptional changes observed in Col-0 plants upon cold exposure were also evident in *ATC* knockdown plants. However, several responses observed in cold-treated Col-0 plants could already be detected in knockdown plants when grown under standard conditions, suggesting that these mutants exhibited typical cold responses without prior cold stimulation. We believe that nucleotide signaling is involved in “cold-like priming” and “cold acclimation” in general. The observed transcript levels of genes involved in central carbon metabolism and respiration were an exception to these findings. These were upregulated in the cold but downregulated in warm-grown *ATC* mutants.

## 1. Introduction

Due to their sessile lifestyle, plants are forced to acclimate changing environmental conditions. Over the last decades, extensive research on various plant species has already led to the discovery of various metabolic pathways important for acclimation of the plant’s metabolism to low temperatures. This complex process of acclimation to low but non-freezing temperatures has been described in plants as “cold acclimation” (CA) and can be divided into three stages involving a variety of molecular, biochemical, and physiological changes [1,2]. In general, CA involves the rearrangement of membrane composition, adjustment of water contents in different tissues, global modification of gene expression, and regulation of protein activity [2,3,4,5].

The first phase of CA involves the direct cellular response to cold, including responses to an energy imbalance between photochemistry, electron transport, and metabolism. State transitions and enhanced non-photochemical quenching can minimize excitation energy at photosystem II (PSII) and therefore reduce the risk of damage [6]. 

To further counteract the effects of low temperatures, in the second phase of cold acclimation, extensive reprogramming of the transcriptome, proteome, and metabolome results in alterations in primary and specialized metabolism [7]. Further, cold treatment often goes along with increased production of reactive oxygen species (ROS) in the plant’s metabolism. Thereby, ROS emerge at various sites in the plant’s metabolism, e.g., at photosystem I, as NADP+, the final acceptor of the photosynthetic electron transport chain, is limited under cold conditions (acceptor-site limitation). To ensure ROS homeostasis, the accumulation of ROS-scavenging compounds is induced, and photosynthetic efficiency is reduced under cold stress [8]. 

In the third and final phase of CA, expression changes that occurred during the second phase of acclimation are established under persistently altered environmental conditions [3]. 

Overall, there is no unique metabolite, transcript, or pathway that can be assigned as responsible for cold tolerance; rather, a multifaceted reorganization of metabolic homeostasis appears to be required. However, different metabolites were shown to have protective effects against different forms of cold stress. 

The accumulation of flavonoids, for example, is of particular importance for the protection against the high reducing ability of the photosynthetic electron transport chain under cold conditions [9]. In the first step of the flavonoid-biosynthesis phenylalanine, derived from the chloroplast, is converted to p-coumaroyl-CoA in the phenylpropanoid pathway in the cytosol. Further reaction steps, which include early (e.g., CHI) and late (e.g., LDOX) biosynthetic enzymes, end with the accumulation of colored anthocyanin pigments in the vacuole. In plants, the regulation of flavonoids is still not completely understood, but several signals have been reported to trigger the induction of the early biosynthetic genes that provide flavonols as well as the late biosynthetic genes leading to anthocyanins as their product [10]. Such signals include sugars, phytohormones, ROS, and transcription factors (e.g., MYB111, TT8, and PAP1) [10,11,12,13,14]. Moreover, transcription factors in general have dominant roles in cold acclimation [15]. 

An interesting phenomenon in CA is known as “cold priming”, where a short cold stimulus induces the induction of a core set of stress response genes, thereby affecting the cold response of plants upon longer exposition to cold [16,17]. 

As cold acclimation requires extensive restructuring of metabolic processes, ribosome biogenesis and translation have to be affected by cold treatment. Indeed, transcriptomic analysis of Arabidopsis during a time course of cold acclimation and deacclimation revealed ribosome biogenesis and translation as key processes involved in cold acclimation [7]. Given that up to 50% of the cellular nucleotides are bound in ribosomal RNA, it becomes clear that sufficient nucleotide provision is a prerequisite for full functionality of the cold acclimation response [18]. 

The enzyme aspartate transcarbamoylase (*ATC*) catalyzes the first committed step in pyrimidine de novo synthesis, and the amount of *ATC* protein is a limiting factor of plant growth [19,20]. It was shown that *ATC* knockdown mutants harbor massively reduced levels of pyrimidine nucleotides (UMP, UDP) and the corresponding nucleotide sugars (e.g., UDP-Glc). This nucleotide limitation and imbalance affected gene expression and photosynthetic performance in *ATC* mutants. Genes involved in central carbon metabolism, intracellular transport, and respiration were mainly downregulated in *ATC* mutants, thereby being putatively responsible for the observed impaired growth. In this work, we aim to unravel and link phenotypical, physiological, and transcriptomic changes in *ATC* knockdown mutants grown in cold temperatures.

## 2. Results

### 2.1. Cold Leads to Increased Expression of Genes Involved in Nucleotide De Novo Synthesis

Although the precise balancing of nucleotides and de novo biosynthesis play an important role in plant development, little research on the role of this metabolic pathway in the abiotic stress response has been performed in the past. Nevertheless, a recent study showed that the biosynthesis of pyrimidines and purines was greatly increased during cold acclimation, resulting in significantly larger nucleotide pools [21]. In-depth analysis of a previously published transcriptome dataset obtained from cold-treated wild-type (Col-0) plants revealed that major transcriptional changes in pyrimidine and purine biosynthesis occur during the early phases of cold acclimation (1–3 days) (Figure 1A,B) [7]. Among all analyzed pyrimidine de novo synthesis genes, dihydroorotate dehydrogenase (DHODH) exhibited the greatest increase in expression (1.8-fold) during cold treatment. Further transcriptional changes could be observed for the plastid-localized enzymes carbamoyl phosphate synthase (CPS SSU/LSU) and aspartate transcarbamoylase (ATC, red) (Figure 1B). In line with this, genes of purine biosynthesis also showed an increase in transcript levels. The most pronounced change was seen in inosine monophosphate dehydrogenase 1 (IMPDH1) (Figure 1A,B). During the de-acclimation phase, transcript levels for both pyrimidine and purine de novo synthesis genes decreased to or even below the initial levels before cold acclimation (Figure 1B).

### 2.2. ATC Abundancy Is Strongly Increased Up on Cold Acclimation and Adaptation

To confirm the increased expression of *ATC* in response to cold in our laboratory, corresponding analyses were carried out using targeted gene expression analysis (qRT-PCR). Plants were grown under control conditions (22 °C, 14 h light/10 h dark) for 21 days and subsequently transferred to 4 °C for defined timespans. The greatest changes in *ATC* transcript levels were seen after 2 and 3 days of cold treatment, with increases up to 3.5-fold compared to the onset of cold treatment (Figure 1C). During prolonged cold exposure (days 8 and 63), the transcript levels again decreased significantly to the initial level before the cold treatment (Figure 1C). Comparable results could be obtained by quantifying protein levels using an *ATC* antiserum raised against the recombinant protein [19]. Thereby, Arabidopsis wild type (Col-0) plants revealed an almost 5-fold increase in *ATC* protein levels after 3 days of cold treatment, which declined to initial levels at 8 days after transfer to 4 °C (Figure 1D). 

After seven days at 4 °C, Col-0 plants showed 40.8% less fresh weight compared to their growth at ambient temperature. In contrast, the fresh weight of the *ATC* mutants was not impaired after 7 days of incubation in the cold, compared to untreated plants of the same genotype (Figure 2A). However, it became obvious that *atc#1* and *atc#2* failed to accumulate anthocyanins (Figure 2B,D). While Col-0 plants accumulated 147-fold more anthocyanins in the cold, the *atc#2* line only showed a 3.6-fold increase compared to the control plants (Figure 2D). Col-0 plants that were exposed to 4 °C for 63 days showed a significant reduction in their maximum growth height, but no differences in plant height could be observed in *ATC* mutants between the control and cold conditions (Figure 2C). It was also found that the chlorophyll content in the *ATC* mutants was reduced, thereby explaining the slightly pale green appearance of the mutant leaves after cold treatment (Figure 2E).

### 2.3. Cold Acclimation Imposes Global Adaptations of the Transcriptome

To investigate the impact of an altered nucleotide metabolism on the transcriptional changes induced during cold acclimation, RNA-Seq analysis was performed. To this end Col-0 and *atc#1* grown at ambient temperature for six weeks (control) were transferred to cold (4 °C) for one and three days, before harvesting the aboveground tissue for RNA isolation. A comparison of transcriptional changes between Col-0 and *atc#1* (grown under ambient temperature) was shown previously [20]. Where necessary for a better understanding, we will also refer to these results in the following.

Overall, when analyzing the differentially expressed genes (DEGs) that were significantly (p_adj._ < 0.05) altered by cold treatment and not filtered by a certain fold change threshold, massive changes in the global transcriptome could be observed in both genotypes. 

Compared to the control conditions, one day of cold treatment resulted in 12,564 DEGs in Col-0 (Figure 3A, Appendix A), and the number of DEGs decreased to 8608 transcripts after 3 days compared to day 1 at 4 °C (Figure 3B, Appendix A). For *atc#1*, 10,878 genes with altered expression were observed after 1 day of cold compared to the control at 22 °C (Figure 3C), and 7603 after three days of cold compared to the same genotype after one day of cold (Figure 3D). Though there was a significant overlap of DEGs between Col-0 and *atc#1* after one to three days of cold treatment, approximately 13% of the DEGs observed for Col-0 were not altered in expression in *atc#1*.

Gene ontology analysis showed that similar groups of transcripts were cold-responsive in both genotypes. After one day of cold, functional groups of DEGs exhibiting the highest significance values were “ribonucleotide-complex biogenesis,” “response to cold,” “response to cadmium ions,” and “photosynthesis” (Figure 3E,F). These are also typical categories found previously in the analysis of plant cold acclimation [7]. When comparing *atc#1* to Col-0 grown at 22 °C, most GO terms differ, with only “photosynthesis” and “response to metal ions” overlapping in both lines [20].

After three days of cold, new categories appear among the ten most significantly regulated functional groups, with “response to water,” ”response to bacterium,” and “response to light intensity” in both genotypes except for “response to bacterium,” which only appeared in *atc#1* (Figure 3G,H). To sum up, for *atc#1*, we found a lower number of DEGs, but in general, changes were part of the same pathways when comparing both genotypes. After 1 day of cold, both genotypes exhibited more upregulated and less downregulated genes, whereas after 3 days of cold, the majority of transcripts were downregulated compared to 1 day of cold in both genotypes. 

A category only appearing in the top 10 list in *atc#1* (one day and three days of cold) is “response to Karrikin,” a compound synthesized during wildfire. Upregulated genes in this category belong to sugar metabolism, photosynthesis, chaperones, and transcription factors. 

When comparing Col-0 to *atc#1* at 22 °C, a total of 2757 DEGs were found. After 1 day of cold, 9303 genes showed differential expression in both Col-0 and *atc#1*, whereas 3261 were only found in Col-0 and 1575 only in *atc#1* (Figure 4A, middle panel). Up- and downregulated genes (red and blue arrows) were similar in numbers, with slightly more upregulation in the intercept but slightly more downregulation in the individual groups (Figure 4A, middle panel). After 3 days of cold, the number of DEGs in the intercept was reduced to 59%, whereas the number of DEGs in the *atc#1* group increased by 37% and was constant in Col-0 (Figure 4A, right panel). When focusing on those DEGs found in both comparisons (1 day of cold and 3 days of cold), the higher number of upregulated genes after one day of cold is reversed after 3 days of cold. In fact, as can be seen from the analysis of the functional groups, genes upregulated after 1 day of cold were often downregulated after 3 days of cold and vice versa. In other words, the global expression after 3 days of a cold is more like the situation before the onset of a cold. 

### 2.4. Transcriptional Response in Functional Groups “Ribosomal Protein” and “Transcription Factor”

A new hub in acclimation to cold was identified with “Translation”. In this category, a high number of ribosomal proteins dominate, and most were found to be upregulated upon cold exposure [7]. From 207 genes encoding ribosomal proteins altered in expression, we identified 154 in Col-0 after one day in the cold but only 135 genes in *atc#1* (Appendix A) [7]. Among these, 13 genes were only altered (upregulated) in *atc#1,* whereas 32 genes were found altered (upregulated) exclusively in Col-0. This could indicate a reduced cold responsiveness of the *atc#1*. In several cases, we observed that the cold response was already seen in *atc#1* grown at ambient temperature, compared to Col-0 (Figure 4B). This phenomenon is similar to “cold priming” [17], but without the requirement of a brief cold stimulus. We also observe this phenomenon in genes involved in nucleotide metabolism (Table 1) and, to a lesser extent, intracellular transport (Table 2), whereas in genes involved in carbohydrate metabolism and respiration for *atc#1,* contrasting expression in ambiently grown plants compared to cold-grown plants was observed (Table 3). 

Because of the clearly defined role of transcription factors (TF) in cold acclimation [15], we inspected the expression of corresponding TF in our experimental setup. (Appendix A). With a focus on those TF showing changes between *atc#1* and Col-0 already at 22 °C (Figure 4C), several TF with massively altered expression upon cold were identified. Among these are BHLH100 (At2g41240) involved in iron homeostasis and leaf growth [22,23], DREB2C (at2g40340) of the DREBA-2 subfamily, involved in drought response [24]; COR47 (At1g20440) involved in cryoprotection [25]; COR314 (At1g29395), cold regulated protein of the chloroplast inner envelope [26]; MYB77 (At3g50060) involved in light dependent auxin signaling [27] and WRKY30 (At5g24110), and WRKY58 (At3g01080) [28].

### 2.5. Transcriptional Response in Functional Groups “Nucleotide Metabolism”, Intracellular Transport” and “Central Carbohydrate Metabolism and Respiration”

For ribosomal proteins and TF, we observed that changes after 1 day of cold were reversed almost completely after 3 days of cold (Figure 4B,C). This was also observed in the next category, “nucleotide metabolism”.

Nucleotide metabolism is directly targeted by knocking down *ATC* (encoded by PYRB). Most genes in de novo synthesis and salvage were upregulated after 1 day cold and partially this is already seen in the comparison of *atc#1* against Col-0 at 22 °C (AIR carb., GMPS, IMPDH1, DNK) (Table 1). Nucleotide catabolic genes (UAH, ALN, PYD2, PYD3) show downregulation, as already seen in *atc#1* grown at ambient temperature compared to Col-0. Furthermore, the pastidic uracil importer PLUTO is upregulated in cold conditions, whereas the nucleotide transporters PMANT1 and ENT3 are downregulated (Table 1).

When inspecting further, intracellular transporters [29] and tonoplast transporters marked the top five downregulated genes (Table 2). Tip1.2 and Tip2.1 function as aquaporins; CLCa is a nitrate transporter; TST1 is a monosaccharide transporter; and CAT2 is a basic amino acid transporter. Furthermore, the tonoplast-located transporters ALMT9 and TDT, both responsible for malate transport, and TST2, a sucrose/monosaccharide transporter, were upregulated in Col-0 and in *atc#1* upon cold (Table 2). In contrast, two plastid-localized carriers were down-regulated in both genotypes (pGlcT, a glucose transporter, and FOLT1, a folate transporter). Strongest induction after 1 day of cold exposure was found for the glucose-6-phosphate-phosphate transporter GPT2 (1d 4 °C/22 °C = log2fc of 5.9 and 7.5 in Col-0 and *act#1*, respectively) [30]. 

In a comparison of DEG between *atc#1* and Col-0 in the warm, a downregulation of glycolysis, tricarboxyclic acid (TCA) cycle and respiration was observed [20]. When inspecting the same set of genes for DEGs in cold-grown plants, 11 genes were found upregulated and only four downregulated in Col-0 and *atc#1* (Table 3). Among those upregulated were AOX1A (alternative oxidase) and NDA1 (alternative NAD(P)H dehydrogenase) both already upregulated in *atc#1* at 22 °C and less responsive in *atc#1* in the cold. Both corresponding gene products function by avoiding mitochondrial respiration’s energy-saving reactions. AOX1A has a function in acclimation to low temperatures and can attenuate ROS production [31,32], a limitation Of ROS production was also observed in NDA1 overexpressor lines [33]. Cytosolic fumarase 2 (*FUM2*) was downregulated in Col-0 after 1 day of cold exposure and further repressed after 3 days of cold exposure (Table 3). Interestingly, *atc#1* already shows downregulation of *FUM2* at 22 °C, and most likely as a consequence, less downregulation after 1 day of cold and no further response after 3 days of cold. Again, this is another example of apparent cold priming at 22 °C. 

### 2.6. ATC Knockdown Showed Diminished Activation of Flavonoid Biosynthesis in the Cold

As we observed reduced anthocyanin levels in cold-grown *ATC* knockdown mutants (Figure 2), we had a closer look at the corresponding genes in the pathway. We observed reduced expression of early biosynthetic genes and a corresponding TF (e.g., *MYB111*) after 3 days of cold treatment (Figure 5A–C). However, a more pronounced effect was observed for the late biosynthetic genes involved in anthocyanin biosynthesis. While strongly induced in Col-0 at 4 °C, *DFR*, *LDOX*, *UGT79B1*,** and the TF *TT8* were markedly reduced in expression in *atc#1* after 3 days of cold (Figure 5D,E).

### 2.7. Alterations in Photosynthetic Rate Led to ROS Accumulation & Reduced Sugar Accumulation in atc#1 and 2

*ATC* knockdown mutants are characterized by impaired photosynthesis [20]. When plants were grown for 28 days in the cold, an increase in photosynthetic efficiency (YII) compared to 22 °C was observed in Col-0; however, this increase was more pronounced in *atc#1* and *atc#2* (Figure 6A). For easier comparison, we only show results for PAR 134; full light curves are given in Appendix A. At the same time, reductions in NPQ levels were detected in all genotypes, again more pronounced in both *ATC* mutants (Figure 6B, Appendix A). Reduced photosynthetic efficiency was accompanied by increased levels of reactive oxygen species, as revealed by NBT and DAB staining (Figure 6C). Surprisingly, levels of ascorbate and dehydro-ascorbate were massively reduced in *atc#2* (Figure 6D). Accumulation of the neutral sugars glucose, fructose, and sucrose is a typical response during cold acclimation, which was corroborated with our analysis of Col-0 exposed to the cold for 28 days (Figure 6E). However, both *ATC* knockdown mutants entirely failed to accumulate sugars in the cold and essentially showed the same carbohydrate level at 4 °C and 22 °C.

## 3. Discussion

Here we showed that upon cold exposure, transcripts encoding enzymes involved in purine and pyrimidine de novo synthesis were induced, while those involved in catabolism were repressed on a global scale (Figure 1). This finding was further corroborated by the transcriptional induction of aspartate transcarbamoylase (*ATC*) in qPCR analysis and the accumulation of *ATC* protein in Col-0 (wild type) plants after 3 days of growth at 4 °C (Figure 1). In line with these observations, upregulation of purine and pyrimidine de novo synthesis was observed by increased amounts of pathway intermediates in a cold-time course experiment as well [21].

These observations agree well with a general upregulation of components for translation (synthesis of ribosomal proteins and ribosome assembly) upon cold exposure, facilitating reprogramming of metabolism during cold acclimation [7]. As ribosomes consist of proteins and RNA, the latter of which is built from nucleotides, it is conceivable that increased translation, requiring ribosome biosynthesis, is demanding increased nucleotide provision. Interestingly, the target of rapamycin (TOR), a central growth regulator, was shown to activate nucleotide de novo synthesis upon growth signals, and vice versa, nucleotide limitation impairs TOR activity [23]. Together with the observation of markedly altered expression of transcription factors involved in the abiotic stress (including cold) response, it becomes likely that nucleotide signaling is at least partially responsible for the massive alterations in gene expression between Col-0 and atc#1 in warm and cold.

Downregulation of the first committed step in pyrimidine de novo synthesis in plants, catalyzed by ATC, leads to impaired growth and development [19,34,35]. Reduced chlorophyll contents, impaired photosynthetic activity, and accumulation of reactive oxygen species (ROS) are very likely causative of the growth phenotype [20]. *ATC* knockdown mutants *atc#1* and *atc#2*, exhibiting a 3- and 20-fold reduction in *ATC* protein abundance compared to Col-0 [19], were unable to accumulate sugars and anthocyanins as usually observed in plants undergoing cold acclimation [13,36] (Figure 2 and Figure 6). Surprisingly, photosynthetic yield (YII) in *atc#1* and *2* was improved in the cold, whereas this was hardly observed in Col-0. However, *ATC* knockdowns still perform worse than controls. When looking at central carbohydrate metabolism, genes upregulated upon cold in both genotypes were downregulated in *atc#1* vs. Col-0 at ambient temperature (Table 3). This may hinder cold acclimation and explain why *ATC* mutants are not able to accumulate neutral sugars in the cold (Figure 6), which is most likely related to the inability of *atc#1* to accumulate anthocyanins (Figure 2) [14]. Similar to carbohydrate metabolism, reduced expression of genes active in anthocyanin metabolism was also observed. A downregulation of early biosynthetic genes CHS and CHI as well as MYB111 transcription factor but even more pronounced all late biosynthetic genes and the TT8 transcription factor was observed in the mutants (Figure 5), explaining the low anthocyanin content in cold treated *ATC* knockdown plants.

In contrast, increased expression of genes encoding ribosomal proteins was observed in *atc#1* plants grown at ambient temperature. When these plants were cold treated, the same set of genes was induced, however, at a lower level as compared to Col-0 at day 1 of cold treatment. For transcripts corresponding to “nucleotide metabolism,” the sum of changes in *atc#1* vs. Col-0 and *atc#1* 4 °C vs. 22 °C equals the changes observed in Col-0 4 °C vs. 22 °C (i.e., the first and third columns of Table 1 with respect to the second columns (e.g., ADSL, AIRCAR, GMPS, and so on). This behavior in a way resembles a phenomenon known as “cold priming,” where a short cold treatment, normally too short to permit the full induction of the acclimation response, affects the expression of a certain set of genes essential to cope with low growth temperatures. Among these are the ROS markers ZAT10 and BAP1 [37]. These were also found upregulated in warm-grown *atc#1* [19]. As a result, we conclude that *atc#1* and the current nucleotide imbalance mimic cold priming via an as-yet unknown mechanism. 

Two further putative scenarios arise from the transcriptomic data: 1. reprogramming to support chloroplast metabolism; and 2. actions to remove excess reducing power and thus limit ROS accumulation. 

Upregulation of TST2 upon cold allows for increased import of sucrose into the vacuole, which represents the main site of sucrose cleavage by invertase [38]. Thus, sucrose and monosaccharides can accumulate in the vacuole, a typical response to cold acclimation also seen in Col-0 in our study (Figure 6). Alternatively, glucose might be exported from the vacuole by monosaccharide transporters [39], to allow for hexose-phosphate synthesis and subsequent reimport into the chloroplast by GPT2, which was highly upregulated in the cold in Col-0 but even more in *atc#1* (Table 2). This apparent futile cycle of chloroplast triose phosphate export and reimport in the form of glucose-6-phosphate could prevent the depletion of Calvin cycle intermediates. Moreover, strongly impaired photosynthesis in *atc#1* means less energy (ATP and reducing equivalents) for growth and development. In line with this observation, in addition to what was seen in Col-0, NTT was already upregulated at 22 °C in *atc#1* and could allow for ATP import into the chloroplast to replenish the energy pool. Reimport of sugars is further supported by reduced expression of TCA cycle genes in *atc#1*, which may further explain the inability of *atc#1* and *2* to accumulate sugars in the cold.

ROS accumulating in *atc#1* and *2* under warm and cold conditions (Figure 6C) may drive the removal of reducing power by the malate valve, converting NADPH and oxaloacetate to malate in the chloroplast, and subsequently exporting this metabolite to the cytosol [31]. We observed higher malate contents in *atc#2* at 22 °C [20], together with increased alternative oxidases AOXA1 and NADPH dehydrogenase ND1, and both can dissipate excess energy without producing more ATP [40]. Under cold conditions, these effects are more pronounced as DIC transporters mediating the exchange of dicarboxylates between mitochondria and cytosol are upregulated, as well as tonoplastidic TDT and ALMT carriers that might balance malate levels between the cytosol and vacuole (Table 2). In addition, AOX1a shows a further marked increase in expression upon exposure to cold. Downregulation of cytosolic fumarase 2 (FUM2) represents a typical cold response [41] and is seen in both genotypes (Table 3). In addition, this response of FUM2 is seen in *atc#1* grown at ambient temperature, together with reduced levels of succinate and fumarate but wild-type-like levels of 2-oxoglutarate [20]. This may point to an export of oxoglutarate to the chloroplast to serve as the carbon skeleton for amino acid synthesis by refixing ammonia released by photorespiration [42,43]. High glyoxylate levels are indicative of high photorespiration, as glyoxylate is a key intermediate of this pathway [43]. In addition, glyoxylate is a product of purine catabolism [44]. Together with the observed accumulation of other intermediates in purine catabolism, this points to purine breakdown to balance purine to pyrimidine pools, latter being low in *atc#1* [20]. 

When oxoglutarate is exported from mitochondria, resulting in low succinate and fumarate, as discussed above, this will lead to low malate generation in the TCA cycle. At the same time, this might support malate production from oxaloacetate in the cytosol, thereby reducing NADH levels.

## 4. Materials and Methods

### 4.1. Plant Growth

For RNA isolation, tissue collection, and phenotypic inspection, wild-type (Col-0) and transgenic *Arabidopsis thaliana* (L.) Heynh. plants (ecotype Columbia) were used throughout. Plants were grown on standardized ED73 soil (Einheitserde and Humuswerke Patzer) under a short day regime (10 h light and 14 h darkness). Light conditions were set to an intensity of 120 μmol quanta m^−2^s^−1^ Illumination was done with LED light (Valoya NS1, Valoya, Finland). The ambient temperature was either set to 22 °C or, for cold treatment, to 4 °C with a constant humidity of 60%. If not stated otherwise, plant material was harvested in the middle of the light period and frozen in liquid nitrogen for further use. The used *ATC* ami-RNA knockdown lines were described previously [19].

### 4.2. RNA Extraction

RNA was extracted from the entire leaf material of soil-grown plants, which was homogenized in liquid nitrogen prior to the extraction of RNA with the Nucleospin RNA Plant Kit (Macherey-Nagel, Düren, Germany) according to the manufacturer’s advice. RNA purity and concentration were quantified using a NanoPhotometer (Implen, Munich, Germany). 

### 4.3. cDNA Synthesis and Gene Expression Analyses

Total RNA was transcribed into cDNA using the qScript cDNA Synthesis Kit (Quantabio, Beverly, MA, USA). QPCR was performed using the quantabio SYBR green quantification kit (Quantabio) on the PFX96 system (BioRad, Hercules, CA, USA) using specific primers. Actin (At3g18780) was used as a reference gene for transcript normalization. All used primers are listed in Appendix A.

### 4.4. RNA Seq and Transcriptome Analyses

All leaf material from at least three plants was sampled for one biological replicate. All conditions were analyzed with three biological replicates. RNA library preparation and transcriptome sequencing were performed by Novogene (Beijing, China). Before library preparation, the isolated RNA was first analyzed for quality by agarose gel electrophoresis and the Agilent 2100 Bioanalyzer (Agilent, Palo Alto, CA, USA). To remove ribosomal RNA and generate the cDNA libraries, poly(A)+ RNA enrichment and mRNA fragmentation followed by random-prime cDNA synthesis was performed. This was followed by paired-end sequencing of the reads of 150 base pairs in length, which were captured using an Illumina sequencer (NovaSeq 6000). To verify and quantify the quality of the reads, the program Hisat2 was used in bioinformatic analyses, and the reads were mapped to the Arabidopsis reference genome. The further bioinformatic evaluation was also performed by Novogene. To investigate which biological processes were highly altered in expression, DESeq2 [45] was used. Therefore, all genes with an adjusted *p*-value < 0.05 are included. We compared the following groups: Col-0 1d_4 °C vs. Col-0_22 °C; Col-0 3d_4 °C vs. Col-0_1d_4 °C; *atc#1* 1d_4 °C vs. *atc#1* _22 °C; *atc#1* 3d_4 °C vs. *atc#1* _1d_4 °C. In the publication Bellin et al. (2021b), we compared *atc#1_*22 °C vs. Col-0_22 °C, and these data were also shown in Figure 4B,C, and Table 1, Table 2 and Table 3. RNA-Seq reads have been uploaded to the Sequence Read Archive (SRA) at NCBI (National Center for Biotechnology Information), BioProject ID: PRJNA925318. FPKM values for all samples are included in Appendix A.

### 4.5. Protein Extraction

Leaf extract of wild-type and knockdown mutants was prepared by homogenizing leaf material in an extraction buffer (50 mM HEPES-KOH, pH 7.2, 5 mM MgCl_2_, 2 mM DTT, 0.01% DDM, 1 mM EDTA, and 2 mM phenylmethylsulfonyl fluoride (PMSF)) on ice. This homogenous extract was centrifuged for 10 min, 20,000× *g* and 4 °C. The supernatant was collected and stored on ice until use. 

### 4.6. Immunoblotting and Western Blot Detection

For immunoblotting 15 µg of a protein extract were separated in a 15% SDS-PAGE gel were transferred onto a nitrocellulose membrane (Whatman, Germany) by blotting. The membrane was blocked in phosphate-buffered saline plus 0.1% [*v*/*v*] Tween 20 (PBS-T) with 3% milk powder for 1 h at room temperature, followed by three washes of 10 min in PBS-T. Then, the membrane was incubated with a rabbit polyclonal antiserum raised against recombinant *ATC* (Eurogentec, Belgium) for 1 h, followed by three washes with PBS-T. Next, the membrane was incubated for 1 h with a horseradish peroxidase (HPR)-conjugated anti-rabbit antibody (Promega, Walldorf, Germany) diluted in PBS-T with 3% milk powder. The result was visualized by chemiluminescence using the ECL Prime Western blotting reagent (GE Healthcare) and a Fusion Solo S6 (Vilber-Lourmat) imager.

### 4.7. Chlorophyll and Anthocyanin Extraction and Quantification

Chlorophyll pigments were extracted from 50 mg of ground leaf tissue in 80% ethanol. After boiling at 95 °C for 10 min and subsequent sedimentation of insoluble contents (5 min, 12,500 rpm), chlorophyll was measured by the absorbance of the supernatant at 652 nm. The calculation was performed as described by [46]. 

Anthocyanin pigments were extracted from 50 mg in an extraction buffer (H_2_O: 1-propanol: HCL (18:1:81)) on ice. After boiling at 95 °C for 3 min while being shaken at 650 rpm, samples were incubated on RT in darkness for 24 h. After centrifugation (15 min, 12,500 rpm), the extinction of the supernatant was measured at 535 nm and 650 nm. Extinction was corrected by Rayleigh’s formula (E_535_-2.2E_650_).

### 4.8. Pulse-Amplitude-Modulation (PAM) Fluorometry Measurements

A MINI-IMAGING-PAM fluorometer (Walz Instruments, Effeltrich, Germany) was used for in vivo chlorophyll measurements. A light curve assays intact, six-week-old plants that were grown for three weeks under standard conditions (22 °C) before being transferred to cold conditions (4 °C). After 10 min of dark adaptation, measurements were performed in a standard light curve setting [47]. Therefore, incrementally increasing light pulses with intensities from PAR (µmol photons m^−2^ s^−1^) 0 to PAR 726 in 14 steps were applied to the plants. 

### 4.9. NBT (O_2_^−^) and DAB (H_2_O_2_) Staining

Superoxide (O_2_^−^) was detected by nitroblue tetrazolium (NBT) staining [48]. For this, leaves from six weeks old plants (three weeks 22 °C and three weeks 4 °C) were used. In brief, leaves were vacuum-infiltrated with 0.1% NBT. 50 mM potassium phosphate buffer (pH 7.8) and 10 mM sodium azide for 20 min and incubated for 1 h at room temperature. Stained leaves were boiled in 95% ethanol for 15 min and photographed. The detection of H_2_O_2_ was performed by treating plants with DAB-HCl. Leaves were vacuum-infiltrated with 5 mM DAB-HCl, pH 3, for 20 min and incubated in the same solution for at least 8 h overnight. Stained leaves were boiled in an ethanol: acetic acid: glycerol (3:1:1) solution under the hood until they turned transparent and were later documented.

### 4.10. Whole-Leaf Ascorbate Determination

Whole-leaf samples from six-week-old plants were harvested six hours after the onset of light. These plants were grown for three weeks under standard conditions (22 °C) and three weeks in cold (4 °C). Ascorbate and dehydroascorbate levels were determined calorimetrically as described previously [49].

### 4.11. Sugar Extraction

For sugar extraction 50 mg of frozen, grounded leaf sample were boiled in 500 µL deionized water for 15 min at 95 °C. After centrifugation (20,000× *g* for 10 min at 4 °C), the supernatant was used for further sugar analysis. Extracted sugars were quantified using a NAD^+^-coupled enzymatic assay [50].

### 4.12. Quantification of Aminoacids

For the quantification of amino acids, the supernatant derived from sugar extraction was used. 20 µL of the probe were mixed with 60 µL 0.2 M pH 8.8 (NaOH) borat and 20 µL 1.28 mM AQC (6-aminoquinolyl-N-hydroxysuccinimidyl carbamate, Synchem UG & Co., KG-S041, dissolved in acetonitrile), and the probe was directly vortexed for 10 s. The measurement was done using high-pressure liquid chromatography (HPLC). 

## Figures and Tables

**Figure 1 molecules-28-01585-f001:**
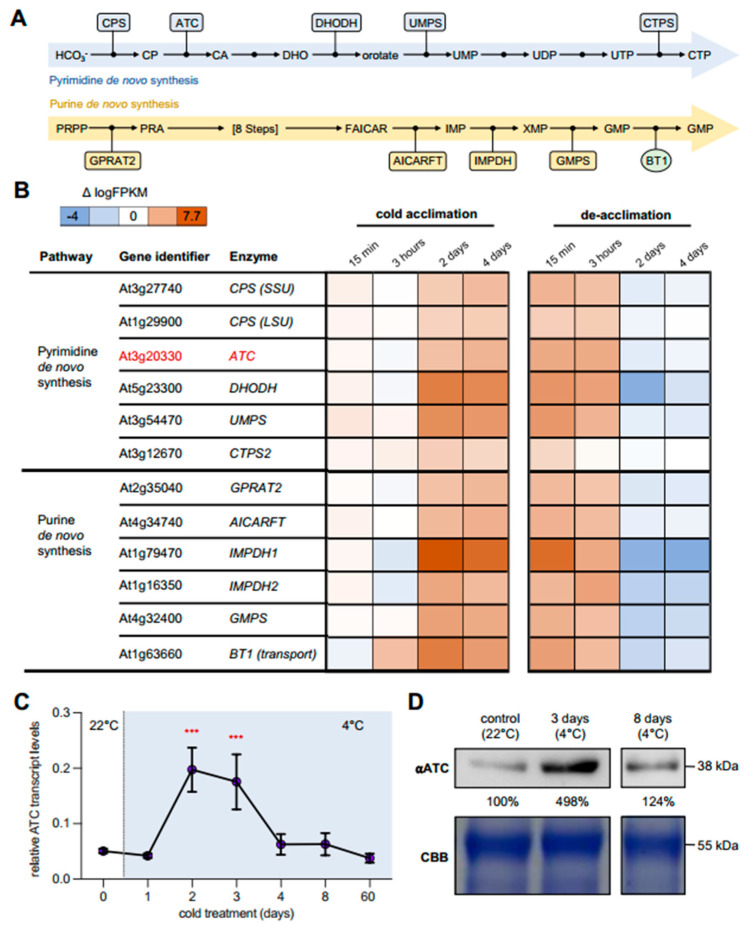
Expression analysis of pyrimidine and purine de novo synthesis genes during cold acclimation and de-acclimation. (**A**) Scheme of enzymes and intermediates of pyrimidine (blue) and purine (yellow) de novo synthesis in plants. (**B**) Expression of purine and pyrimidine de novo synthesis genes in a cold acclimation/deacclimation time course [7]. The main enzyme (*ATC*) studied in this work is indicated in red. Plants were grown for 14 days at ambient temperature before being transferred to 4 °C for four days and immediately transferred to ambient temperature again for an additional four days. Sampling took place at the indicated timepoints. Shown are the Δlog FPKM values compared to 0 min of cold treatment. (**C**) Expression of *ATC* during cold acclimation and adaptation. Col-0 plants were grown at 22 °C for 21 days before cold treatment. Expression was normalized to actin2. Data points are means of three biological replicates +/− standard deviation. (**D**) Similar plants were used for immunoblot staining with *ATC* antibody on whole leaf extracts (top) after 0, 3, and 8 days of cold treatment. Coomassie Brilliant Blue (CBB)-stained SDS-PAGE was used as a loading control. Asterisks depict significant changes relative to the 0 days timepoint according to one-way ANOVA test (*** = *p* < 0.001). Carbamoyl phosphate synthase (CPS) small subunit (SSU) and large subunits (LSU), carbamoyl-phosphate (CP), aspartate transcarbamoylase (*ATC*), carbamoyl-aspartate (CA), dihydroorotate (DHO), dihydroorotate dehydrogenase (DHODH), uridinemonophosphate synthase (UMPS), and cytidintriphosphate synthetase 2 (CTPS2), amidophosphoribosyltransferase (GPRAT2), IMP cyclohydrolase (AICARFT), inosinemonophosphate dehydrogenase1/2 (IMPDH1/IMPDH2), guanosin-monophosphate synthase (GMPS), and brittle transporter 1 (BT1).

**Figure 2 molecules-28-01585-f002:**
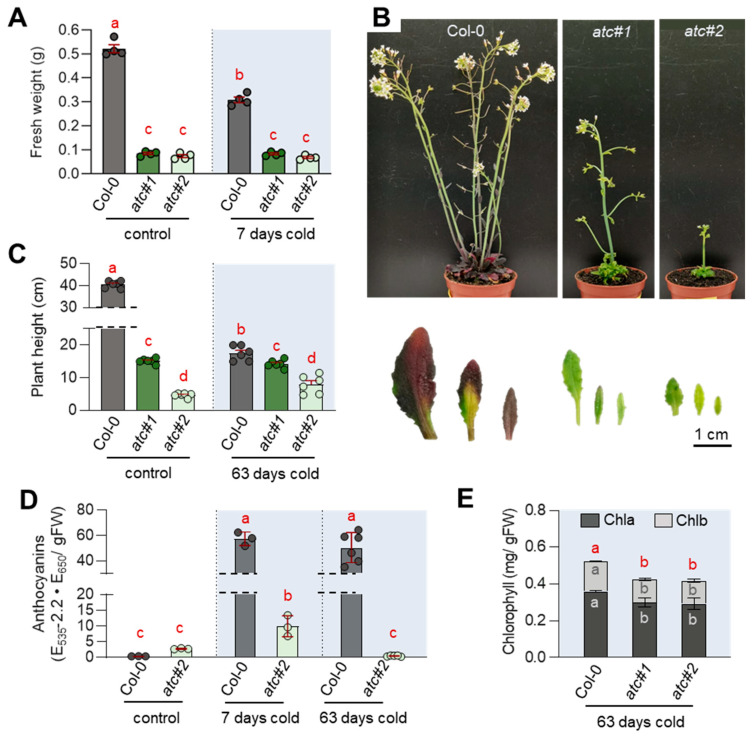
Cold acclimation and adaptation in Col-0 and *ATC* mutants. Plants were grown for three weeks at 22 °C before being transferred to 4 °C for the indicated timespan. Control plants were continuously grown at 22 °C for four weeks. (**A**) Fresh weight (n = 4) (**B**) phenotypical appearance after 63 days cold with closeup of leaves (**C**) plant height (n = 6) (**D**) anthocyanin contents of leaves (n = 3 for control and 7-day-old plants; n = 6 for 63-day-old plants) (**E**) chlorophyll contents (n = 4). Data points and means +/− standard deviation are shown. Different letters denote significant differences according to a two-way ANOVA with post-hoc Tukey HSD testing (*p* < 0.05).

**Figure 3 molecules-28-01585-f003:**
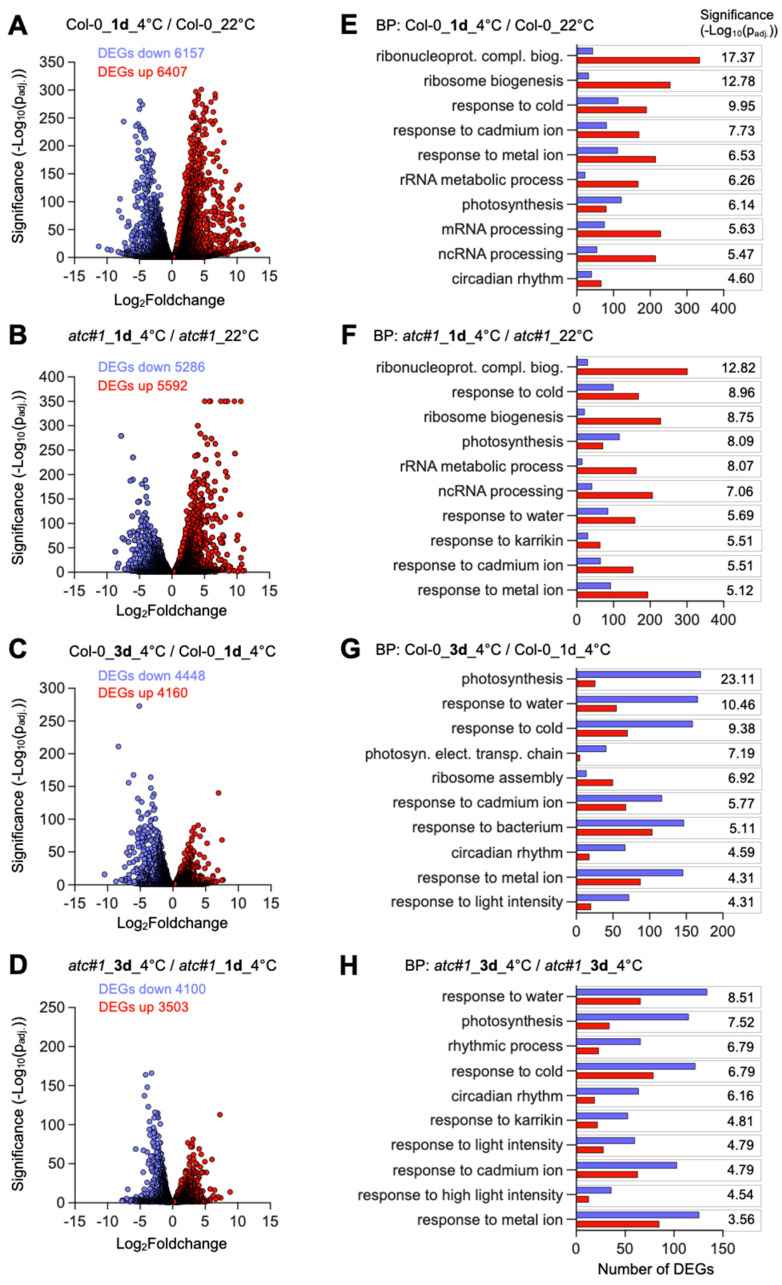
Global transcript alterations in Col-0 and *atc#1* mutants during cold acclimation. Plants were grown for six weeks at 22 °C before being transferred to 4 °C for one or three days. (**A**–**D**) Volcano plots show differentially expressed genes (DEGs) (p_adj._ < 0.05) in Col-0 (**A**,**C**) and *atc#1* mutants (**B**,**D**) comparing one day in the cold to the control (**A**,**B**) and three days of the cold to one day of the cold treatment (**C**,**D**). All listed DEGs were used for (**E**–**H**) GO-Term analysis, revealing significantly altered biological processes (up-regulated in red and down-regulated in blue color).

**Figure 4 molecules-28-01585-f004:**
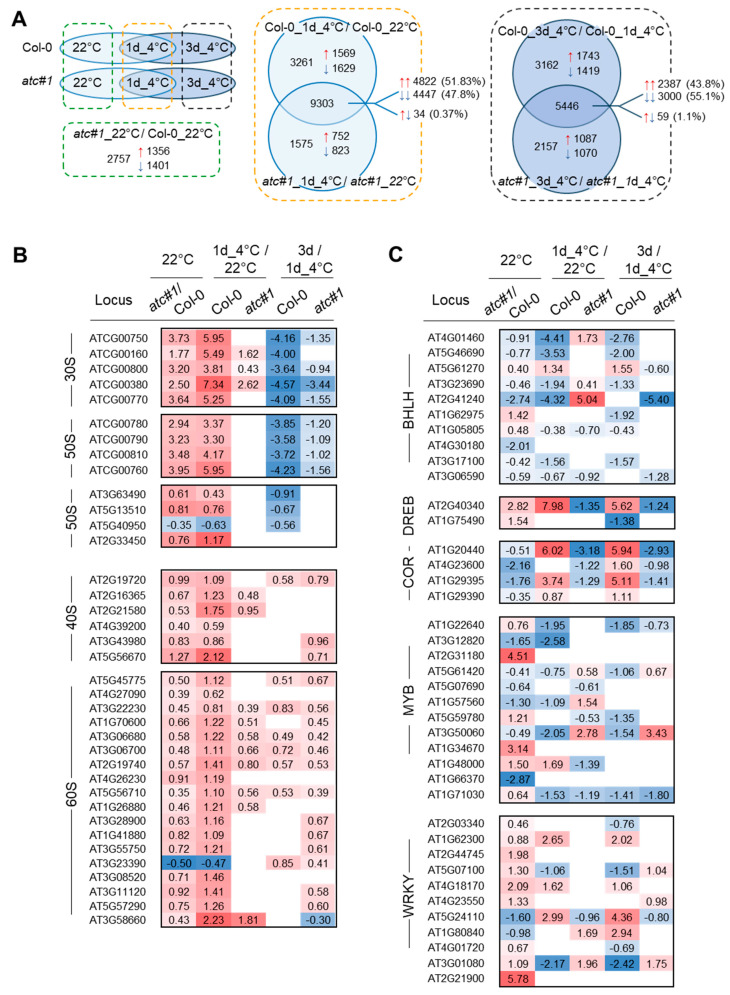
Comparison of global transcript alterations between Col-0 and *atc#1* mutants and assignment to transcription factors and ribosomal proteins during cold acclimation. Plants were grown for six weeks at 22 °C before being transferred to 4 °C for one or three days. (**A**) Visualization of compared datasets (left panel) and Venn diagrams of data comparisons after 1 and 3 days of cold treatment (middle and right panel). (**B**) Ribosomal proteins selected for differential expression at 22 °C (primed) (p_adj._ < 0.05). (**C**) Transcription factors selected for differential expression at 22 °C (primed) (p_adj._ < 0.05) in the blank fields; no significant change. Up-regulated genes are shown in red and down-regulated genes in blue color).

**Figure 5 molecules-28-01585-f005:**
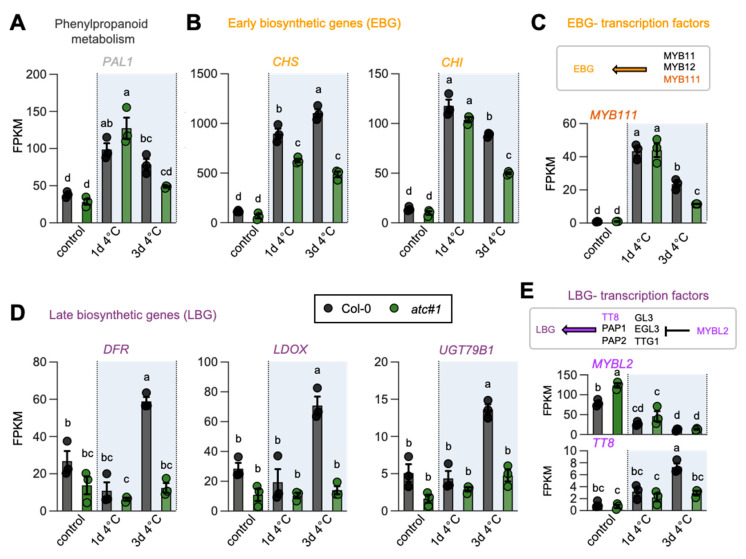
Global transcript alterations (*atc#1* vs. Col-0) in flavonoid biosynthesis genes. Data points and means +/− standard deviation of FPKM values are shown. Different letters denote significant differences according to a two-way ANOVA with post-hoc Tukey HSD testing (*p* < 0.05). Phenylalanine ammonia-lyase (PAL1), chalcone synthase (CHS), chalcone isomerase (CHI), dihydroflavonol-4-reductase (DFR), leucoanthocyanidin dioxygenase (LDOX), cyanidin 3-O-glucoside 2″-O-xylosyltransferase (UGT79B1), *Arabidopsis* MYB transcription factors (MYB111 and MYBL2) and transparent testa8 (TT8).

**Figure 6 molecules-28-01585-f006:**
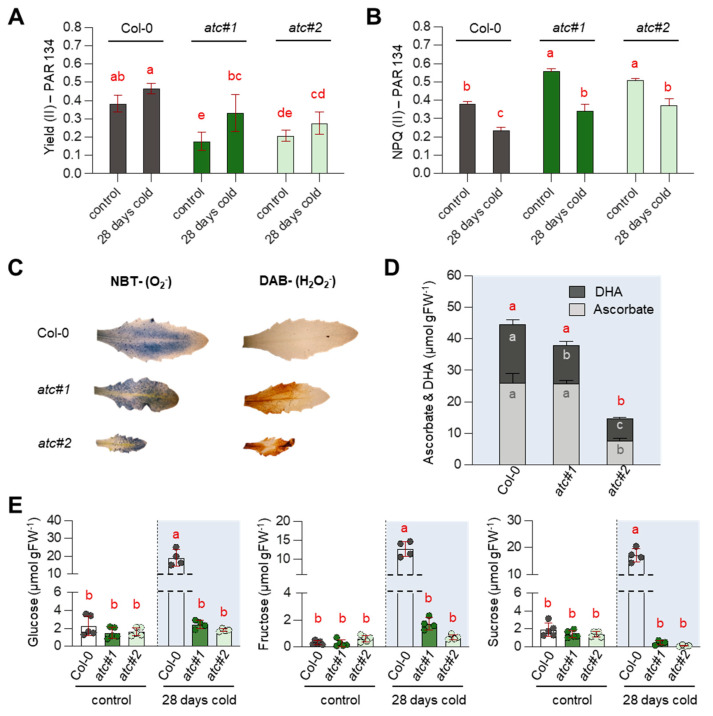
Determination of photosynthetic efficiency, accumulation of ROS, and sugar quantification. Plants were grown for two weeks at 22 °C before being transferred to 4 °C for the indicated times. (**A**) The effective quantum yield of PS (II) and (**B**) non-photochemical quenching (NPQ) were measured in a light response curve (n = 5). (**C**) NBT- and DAB-staining for visualization of reactive oxygen species showing typical results (**D**) measurements of Ascorbate and Dihydroascorbate (n = 5). (**E**) Quantification of glucose, fructose, and sucrose contents (n = 4). Data points are means +/− standard deviation. Different letters denote significant differences according to a two-way ANOVA with post-hoc Tukey HSD testing (*p* < 0.5).

**Table 1 molecules-28-01585-t001:** DEGs involved in nucleotide metabolism. Selected genes (p_adj._ < 0.05) are shown. AGI, Arabidopsis genome identifier.

		22 °C	1 d 4 °C/22 °C		3 d/1 d 4 °C		
Locus	Name	*atc#1*/Col-0	Col-0	*atc#1*	Col-0	*atc#1*	Function
AT4G18440	ADSL	−1.27	−0.98	−0.46	−1.09	−0.50	de novo
AT2G37690	AIR carb.	0.69	1.90	1.36	0.88	0.43	de novo
AT2G37250	AMK1		1.37	1.56		−0.43	de novo
AT5G35170	AMK5		−0.72	−1.14	−1.16	−0.50	de novo
AT3G01820	AMK7		2.97	3.16		−0.80	de novo
AT4G34740	ATase2		2.34	2.34	−0.63	−0.58	de novo
AT5G23300	DHODH		1.62	1.09			de novo
AT1G30820	CTPS1		1.84	2.29	−2.82	−2.31	de novo
AT3G12670	CTPS2		1.95	2.03	−0.99	−0.82	de novo
AT2G16370	DHFR-TS		−1.45	−1.22			de novo
AT3G06200	GMK		1.52	1.05			de novo
AT1G63660	GMPS	0.73	1.53	0.88			de novo
AT1G79470	IMPDH1	0.76	1.80	0.96			de novo
AT1G16350	IMPDH2	0.51	−0.73	−0.92	1.14	0.92	de novo
AT2G21790	RNR1		0.65	1.03	0.52	0.41	de novo
AT3G60180	UMK1		−1.28	−1.39		0.95	de novo
AT4G20070	AAH		2.57	2.26	−1.60	−1.34	catabolism
AT5G43600	UAH		−1.57	−1.42	0.53	0.61	catabolism
AT4G04955	ALN		−2.43	2.20		1.00	catabolism
AT1G05620	NSH2		−1.39	−0.75	1.25	0.75	catabolism
AT3G17810	PYD1	−0.53	0.44	1.00	−1.50	−1.18	catabolism
AT5G12200	PYD2	−0.41	−1.19	−0.90			catabolism
AT5G64370	PYD3	−0.42	−1.17	−0.69			catabolism
AT2G26230	UOX	−0.37	1.01	1.25		−0.53	catabolism
AT1G80050	APT2		−1.58	−1.89		1.02	salvage
AT4G22570	APT3	−0.74	−2.90	−2.66	1.29	1.75	salvage
AT5G11160	APT5		1.13	1.54		−0.59	salvage
AT1G72040	dNK	0.63	1.14	0.65			salvage
AT1G71750	HGPRT		1.06	1.00			salvage
AT5G40870	UCK1		−0.85	−1.04		0.56	salvage
AT5G13490	AAC2		0.97	2.09			transporter
AT5G61810	APC1		2.23	1.88	−1.55	−1.34	transporter
AT4G32400	BT1		2.01	1.58	−0.70	−0.72	transporter
AT4G05120	ENT3		−1.30	−0.69		0.80	transporter
AT4G05110	ENT6				−1.06		transporter
AT5G03555	PLUTO		2.08	2.39	−1.28	−1.29	transporter
AT5G56450	pmANT1		−1.00	−0.77			transporter

**Table 2 molecules-28-01585-t002:** DEGs involved in intracellular transport. Selected genes (p_adj._ < 0.05) are shown. AGI, Arabidopsis genome identifier; Glc6P, glucose 6 phosphate; TP, triose phosphate.

Locus	Name	22 °C	1 d 4 °C/22 °C		3 d/1 d 4 °C		Subcell. Localization	Substrate
		*atc#1*/Col-0	Col-0	*atc#1*	Col-0	*atc#1*		
AT1G14140	UCP3		−1.16	−0.93	0.62		Mt	protons
AT5G58970	UCP2	−0.86	−1.14	−1.10	1.17	1.71	Mt	protons
AT5G09470	DIC3		3.33	4.26			Mt	dicarboxylate
AT4G24570	DIC2		−0.89		3.23	4.04	Mt	dicarboxylate
AT2G22500	DIC1	−0.85	2.15	2.97			Mt	dicarboxylate
AT3G48850	PIC2		1.41	4.35	−1.48	−2.67	Mt	phosphate
AT2G17270	MPT1		2.32	2.42	−1.04	−1.00	Mt	phosphate
AT5G01340	SFC		3.15	3.46		−1.16	Mt	succinate/fumarate
AT5G27520	PNC2		1.97	2.05	−1.48	−1.49	Mt, P	adenine nucleotide
AT5G66380	FOLT1		−0.76	−1.05		0.65	PL	folate
AT5G16150	pGlcT	−0.44	−0.87	−1.03	−1.20	−0.34	PL	hexose
AT1G68570	NPF3.1		2.69	2.35	−0.66	−0.72	PL	nitrite
AT4G32400	BT1-like		2.01	1.58	−0.70	−0.72	PL	nucleotide
AT1G61800	GPT2		5.90	7.61	0.39	−1.65	PL	Glc6P, TP
AT1G58030	CAT2		−1.40	−1.32		0.62	TP	amino acid
AT5G40890	CLCa		−2.34	−1.50		0.56	TP	Cl, nitrate
AT3G16240	TIP2.1		−3.42	−3.74	−0.48	0.97	TP	H_2_O
AT3G26520	TIP1.2		−2.08	−1.85	−0.49	0.39	TP	H_2_O
AT3G18440	ALMT9		1.40	0.92	−0.71	−0.86	TP	malate
AT5G47560	TDT		1.82	0.96	−2.45	−1.47	TP	malate
AT1G20840	TST1	−0.84	−1.50	−1.10	0.89	0.66	TP	monosaccharide
AT4G35300	TST2		2.94	2.61	−1.07	−1.22	TP	mono-, disaccharide

**Table 3 molecules-28-01585-t003:** DEGs involved in central carbohydrate metabolism and respiration. Selected genes (p_adj._ < 0.05) are shown.

Locus	Name	22 °C	1 d 4 °C/22 °C		3 d/1 d 4 °C		Localization	Function
		*atc#1*/Col-0	Col-0	*atc#1*	Col-0	*atc#1*		
AT5G50950	FUM2	−1.49	−2.38	−2.14	−0.74		Mt, Ct	fumarase
AT1G07180	NDA1	0.59	1.07		−0.73		Mt, P	dehydrogenase
AT3G22370	AOX1A	0.70	3.00	2.47	−1.74	−1.97	Mt	reductase
AT3G55410	E1-OGDH1	−0.33	0.99	1.37			Mt	dehydrogenase
AT5G55070	E2-OGDH2	−0.62	1.06	1.63		−0.43	Mt	dehydrogenase
AT4G35260	IDH1	−0.51	0.80	1.04			Mt	dehydrogenase
AT3G27380	SDH2-1	−0.43	1.68	2.22	−1.14	−1.59	Mt	dehydrogenase
AT3G03250	UGP2	−0.57	1.19	0.95	−0.40	−0.52	Ct, PM	pyrophosphorylase
AT5G51830	FRK1	−0.53	2.46	3.27	−1.22	−1.93	Ct	kinase
AT5G56630	PFK7	−0.35	2.15	1.87	−1.26	−1.44	Ct	kinase
AT5G56350	PK	−0.46	1.59	2.04		−0.76	Ct	kinase
AT4G26520	FBA7	0.40	−1.85	−1.62	0.61		Ct	aldolase
AT3G52930	FBA8	−0.38	1.48	1.26	−0.47	−0.49	all	aldolase
AT1G36380	Cyt C red	−0.77	−1.10			−0.61	?	unknown

## Data Availability

RNA-Seq reads have been uploaded to the Sequence Read Archive (SRA) at NCBI (National Center for Biotechnology Information; BioProject ID: PRJNA925318).

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
