# Peer review of "Nucleotide Imbalance, Provoked by Downregulation of Aspartate Transcarbamoylase Impairs Cold Acclimation in Arabidopsis"

_molecules, 2023, doi:10.3390/molecules28041585_

Round 1

Reviewer 1 Report

This manuscript explores mainly the transcriptional changes that occur in plants that are severely compromised in pyrimidine nucleotide biosynthesis (ATC knockdown mutants) upon cold exposure spiced up with some physiological data. The observed changes are drastic, and interestingly the knockdown plants fail to show some typical cold responses, for example anthocyanin production, while some responses seem constitutively induced even before cold stress exposure.

I have no major suggestions for improvement. I just wondered about the proposal in the abstract that nucleotides might contribute to induce signaling for cold acclimation. There are so many changes due to cold and due to the ATC mutation that I find it daring to come to such a concrete idea – does that not particularly depend on which of the many changes one decided to focus on. If I am wrong, the authors could try to explain especially this idea better in the manuscript.

In the discussion, I had problems to follow the line of argument in several instances (see below). Also here it may be necessary to clarify the explanations.

Minor comments:

52, 53. The sentence regarding the ‘third phase’ is difficult to understand. Please rephrase.

93, response

Figure 1B. Although it could be looked up in the given publication, it would be nice to mention here some more experimental details: age of the plants; duration of cold-treatment before de-acclimation, for example.

Figure 1D. The plants were similar / the same? as those used for Fig. 1C?

132, remove hyphen from ‘transcarbamoylase’

156, the failure to accumulate anthocyanins is shown in Fig. 2B and D (not C). Also probably wrong in the next sentence.

Figure 2. The number of measurements should be mentioned, although clear due to the data points shown, at least not clear for E. Also in Fig. 6.

211, the wording makes it difficult to understand: 9303 genes showed differential expression in both Col-0 and atc#1 (not ‘between’),

314-316 please mention that the reduced expression was observed in the mutant

333, both

354, in Chen et al also increased amounts of pyrimidine nucleotides were observed after cold

402-407, the rationale of these sentences is unclear to me. Sugars are exported in the cold from the vacuole by transporters that are downregulated under these conditions?  To explain why GPT2 is upregulated… This is a bit twisted.

411-413, sentence is unclear

414-417, when reducing power is low, why would that activate the removal of reducing power? You may want to be more detailed which cellular compartments you refer to – maybe also which malate valve you mean.

425, typical response for what?

426-427, more glyoxylate, less succinate and fumarate – why does that ‘point to a branch’ in the citric acid cycle?

428, reducing routes of malate production – what does that mean?

Reviewer 2 Report

This manuscript is about the importance of aspartate transcarbamoylase regulation in the cold acclimation of Arabidopsis. The authors explain the responses of the cell (both wild-type and mutants) via transcriptome analyses and carried out some phenotypic analyses as supporting materials. Other than the transcriptome data generated from the current study, the authors reanalyzed another 2 sets of transcriptome data from previous studies. Please refer below for specific comments.

Specific comments:

  1. Since the authors reanalyzed previous transcriptome dataset, it is good to specify clearly in the methodology part, maybe at the end of section 4.4 or as a new section 4.5. Please state the relevant accession number from these previous studies. For the current dataset, the authors mentioned that it will be submitted shortly, but I look forward to see the accession number in the next review as this is an important criteria for the manuscript being accepted for publishing.

  2. There are some typos, please check the whole manuscript. For example, L93 ‘rewponse’, L463 ‘Illumina sequencing’ to ‘Illumina sequencer’, L473, L489, L494 ‘20.000 rpm’ dot should be comma ‘20,000 rpm’, L527-528 ‘0,2M pH 8,8’ should be dot in the values, etc. Double check other values as well.

  3. Comprehensive introduction but reordering of the paragraphs is suggested (L39-57). For example, does paragraph (L47-51) belong to second phase? L71, should cold priming be written before the three phases, maybe at L33?

  4. The authors should clearly state the abbreviation when it was first shown in the text. For example, L42, PSII photosystem II. L118, the abbreviation ‘Col-0’ was first shown in the main text, does it mean the wild-type of Arabidopsis? L439, what is half-strength MS?.

  5. L152 ‘After 4 days …’ should be 7 days? L154, remove ‘although’ and why 21 days of growth? L156, (Figure 2B, C) should be (Figure 2B, D). L158, should be Figure 2D. Double check the rest of the elaboration and its corresponding Figure.

  6. Supplemental Data 1-4 were not available for downloading.

  7. In this study, the transcriptome experiment design consists of multi factors, (a) wild-type vs mutant, (b) untreated (22°C) vs cold-treatment, and (c) 1 day vs 3 days treatment. Due to the complexity of the study, it seems like there are some missing comparison groups (more important group) for DEGs analysis in this study. Any reason that the groups “WT4-3days vs WT4-1day” and “Mutant4-3days vs Mutant4-day” were used in this study rather than “WT4-3days vs WT22” and “Mutant4-3days vs Mutant22”? How about “Mutant4-1day vs WT4-1day” and “Mutant4-3days vs WT4-3days”? The authors should justify the reason for choosing the current way of comparison and are there any better comparison group? Otherwise, a multifactor DEGs analysis should be conducted.

  8. Figure 4B, 4C, Table 1-3. As mentioned in comment #7, please amend accordingly the figures and tables after deciding the comparison groups.

  9. Please specify the detailed DEGs analysis in the methodology part. Abbreviations for the comparison groups of DEGs should be used as it is currently very difficult to read the result.

  10. L255, what does in meant by (primed) in the figure legend?

  11. L432, DNA isolation? L438, which growth experiments were conducted on sterile agar plates and were seed samples used in RNA extraction and other experiments?

  12. L445, for current study, how many leaves samples were subjected to RNA extraction? The authors should clearly state the types of samples being studied in the methodology part. Control (without cold treatment) of both wild-type and mutants, and samples for cold treatment (1 and 3 days) for both wild-type and mutants?

Round 2

Reviewer 2 Report

The authors had revised the manuscript based on previous comments, but there remains minor errors. Please refer below for some errors that are being found so far. As an author, it is hard to spot our own errors or blind spots, therefore I suggest the authors to submit the manuscript for English editing service.

Minor comments:

1. L46 "reactive oxygen species (ROS)"

2. Figure 2A y-axis, "09 06 03" to "9 6 3"

3. L150, 28 days? No 28 days data in Figure 2, please specified if "Data not shown" and put the (Figure 2A) to the previous sentence.

4. L164 remove "respectively"

5. L172 onwards check the format of the values again - 12.564 8.608 10.878 2757 9303 1575.

6. L200 (E-H)

7. L260 salvage are upregulated

8. L314 WT to Col-0

9. L402 (Figure 7)
